# Influence-Directed Explanations for Deep Convolutional Networks

## Abstract

We study the problem of explaining a rich class of behavioral properties of deep neural networks. Our *influence-directed explanations* approach this problem by peering inside the network to identify neurons with high *influence* on the property of interest using an axiomatically justified influence measure, and then providing an *interpretation* for the concepts these neurons represent. We evaluate our approach by training convolutional neural networks on Pubfig, ImageNet, and Diabetic Retinopathy datasets. Our evaluation demonstrates that influence-directed explanations (1) localize features used by the network, (2) isolate features distinguishing related instances, (3) help extract the essence of what the network learned about the class, and (4) assist in debugging misclassifications.

## 1 Introduction

We study the problem of explaining a class of behavioral properties of deep neural networks, with a focus on convolutional neural networks. Examples of such properties include explaining why a network classified an input instance a particular way, why it misclassified an input, and what the essence of a class is for the network. This problem has received significant attention in recent years with the rise of deep networks and associated concerns about their opacity (Knight, 2017).

This paper introduces *influence-directed explanations* for deep networks. It involves peering inside the network to identify neurons with high *influence* and then providing an *interpretation* for the concepts they represent. This approach enables us to interpret the inner workings of the network by drawing attention to concepts learned by the network that had a significant effect on the property that we seek to explain. In contrast to raw inputs, neurons in higher layers represent general concepts. Thus, they form a useful substrate to explain properties of interest involving many input instances, such as the essence of a class. Once influential neurons have been identified, they can be interpeted using existing techniques (e.g., visualization) to reveal the concepts they represent. Alternatively, influences can be examined directly to diagnose undesirable properties of the network.

A key contribution of this paper is *distributional influence*, a measure for internal neurons that is axiomatically justified. Distributional influence is parameterized by a quantity of interest, a distribution of interest, and a slice of the network that allows us to reference some internal neurons in a network. It is simply the average partial derivative with respect to a neuron in a slice over the distribution of interest. This parametric measure can be appropriately instantiated to explain different properties of interest with respect to different parts of a network.

Our influence measure is designed to achieve three natural desiderata: *causality*, *distributional faithfulness*, and *flexibility*. Capturing causality helps us identify parts of the network that when changed have the most effect on outcomes. Distributional faithfulness ensures that we evaluate the network only on points in the input distribution. This property is important since models operating on high dimensional spaces, such as neural networks, are not expected to behave reliably on instances outside the input distribution. Finally, by flexibility, we mean that the influence measure should support explanations for various properties of interest.

We evaluate our approach by training convolutional neural networks on ImageNet (Russakovsky et al., 2015), PubFig (Kumar et al., 2009), and a Diabetic Retinopathy datasets. Our evaluation demonstrates that influence-directed explanations enable us to (1) characterize why inputs were classified a particular way in terms of high-level concepts represented by influential neurons (Section 3.1), (2) explain why an input was classified into a one class (e.g., sports car) rather than another (e.g.,

convertible) (Section 3.2), (3) demonstrate that influences localize the actual reasons used for classification better than simply examining activations (Section 3.3.1), (4) help extract the essence of what the network learned about the class (Section 3.3), and (5) assist in debugging misclassifications of a Diabetic Retinopathy classifier (Pratt et al., 2016) (Section 3.4).

## 1.1 RELATED WORK

Prior work on interpreting CNNs has focused on answering two questions: (i) given an input image, what part of the instance is relevant to a particular neuron? (Simonyan et al., 2014; Zeiler & Fergus, 2014; Springenberg et al., 2015; Ribeiro et al., 2016; Sundararajan et al., 2017), and (ii) what maximizes the activation of a particular neuron? (Simonyan et al., 2014; Girshick et al., 2014).

**Localizing relevance**   One approach to interpreting predictions for convolutional networks is to map activations of neurons back to regions in the input image that are the most relevant to the outcomes of the neurons. Possible approaches for localizing relevance are to (i) visualize gradients (Simonyan et al., 2014; Sundararajan et al., 2017; Bach et al., 2015) (ii) propagate activations back using gradients (Zeiler & Fergus, 2014; Springenberg et al., 2015; Bach et al., 2015), (iii) learning interpretable models predicting the effect of the presence of regions in an image (Ribeiro et al., 2016).

**Maximizing activation**   An orthogonal approach is to visualize features learnt by networks by identifying input instances that maximally activate a neuron, achieved by either optimizing the activation in the input space (Simonyan et al., 2014), or searching for instances in a dataset (Girshick et al., 2014).

Our approach differs from prior work along several axes. First, examining causal influence of neurons rather than their activations (Simonyan et al., 2014; Girshick et al., 2014) better identifies neurons used by a network for classification. Experiments in Section 3.3.1 demonstrate why examining activations fails to identify important neurons. Second, our explanations are parametric in a distribution of interest, allowing us to explain the network behavior at different levels of granularity (e.g., an instance or a particular class). Examining the influence of internal neurons plays an important role here because they capture more general concepts, and we demonstrate it is possible to identify "expert" neurons for certain distributions (Section 3.3.1). In contrast, influences of inputs (pixels) considered in prior work do not generalize well across instances of a population. Third, our explanations are parametric in a quantity of interest that allow us to provide explanations for different behaviors of a system, as opposed to instance outcomes. Finally, our choice of influence measures are guided by axiomatic choices different from those in prior work (Sundararajan et al., 2017). A notable difference stems from our distributional faithfulness criteria, which imposes a weaker distribution marginality principle than the marginality principle imposed by Integrated Gradients. A practical consequence of this choice is that it constrains acceptable baseline images that can be used with Integrated Gradients. In contrast, the prior work does not provide formal guidance on the choice of the baseline (see Appendix D for details).

## 2 INFLUENCE

In this section, we propose *distributional influence*, an axiomatically justified family of measures of influence. Distributional influence is parameterized by a quantity of interest, and a distribution of interest, and is simply the average partial derivative over the distribution of interest. In Section 2.2, we show that this is the only measure that satisfies some natural properties. In Section 2.3, we show how this influence measure can be extended to measure the influence of internal neurons.

## 2.1 A FAMILY OF INFLUENCE MEASURES

We represent quantities of interest of networks as continuous and differentiable functions $f$ from $\mathcal{X} \to \mathbb{R}$, where $\mathcal{X} \subseteq \mathbb{R}^n$, and $n$ is the number of inputs to $f$. A distributional influence measure, denoted by $\chi_i(f, P)$, measures the influence of an input $i$ for a quantity of interest $f$, and a distribution

of interest $P$, where $P$ is a distribution over $\mathcal{X}$:

$$\chi_i(f, P) = \int_{\mathcal{X}} \left.\frac{\partial f}{\partial x_i}\right|_{\mathbf{x}} P(\mathbf{x})d\mathbf{x}. \qquad (1)$$

Our measures are parameterized along two dimensions: a quantity of interest $f$, and a distribution of interest $P$. Examples of quantities of interest are: outcome towards the 'cat' class ($f_{\mathrm{cat}}$), or outcome towards 'cat' versus 'dog' ($f_{\mathrm{cat}} - f_{\mathrm{dog}}$). The first quantity of interest answers the question of why a particular input was classified as a cat, whereas the second can be helpful in debugging why a 'dog' instance was classified as a 'cat'.

Examples of distributions of interest are: (i) a single instance (influence measure just reduces to the gradient at the point) (ii) the distribution of 'cat' images, or (iii) the overall distribution of images. While the first distribution of interest focuses on why a single instance was classified a particular way, the second explains the essence of a class, and the third identifies generally influential neurons over the entire population. A fourth instance is the uniform distribution on the line segment of scaled instances between an instance and a baseline, which yields a measure described in Sundararajan et al. (2017) called Integrated Gradients.

## 2.2 Axiomatic treatment

We narrow the space of influence measures using three axioms that characterize desirable properties of influence measures for machine learning models with respect to a quantity and distribution of interest, and then prove that these axioms uniquely define the above measure.

The first axiom, *linear agreement* states that for linear systems, the coefficient of an input is its influence. Measuring influence in linear models is straightforward since a unit change in an input corresponds to a change in the output given by the coefficient.

**Axiom 1** (Linear Agreement). *For linear models of the form* $f(\mathbf{x}) = \sum_i \alpha_i x_i$, $\chi_i(f, P) = \alpha_i$.

The second axiom, *distributional marginality* states that gradients at points outside the support of the distribution of interest should not affect the influence of an input. This axiom ensures that influence measure only depends on the behavior of the model on points within the manifold containing the input distribution.

**Axiom 2** (Distributional marginality (DM)). *If,* $P(\left.\frac{\partial f_1}{\partial x_i}\right|_X = \left.\frac{\partial f_2}{\partial x_i}\right|_X) = 1$, *where $X$ is the random variable over instances from $\mathcal{X}$, then* $\chi_i(f_1, P) = \chi_i(f_2, P)$.

The third axiom, *distribution linearity* states that the influence measure is linear in the distribution of interest. This ensures that influence measures are properly weighted over the input space, i.e., influence on infrequent regions of the input space receive lesser weight in the influence measure as compared to more frequent regions.

**Axiom 3** (Distribution linearity (DL)). *For a family of distributions indexed by some $a \in \mathcal{A}$,* $P(x) = \int_{\mathcal{A}} g(a)P_a(x)da$, *then* $\chi_i(f, P) = \int_{\mathcal{A}} g(a)\chi_i(f, P_a)da$.

We can show that the only influence measure that satisfies these three axioms is the weighted gradient of the input probability distribution (see Appendix B for the proof).

**Theorem 1.** *The only measure that satisfies linear agreement, distributional marginality and distribution linearity is given by Equation 1.*

## 2.3 Internal influence

In this section, we generalize the above measure of input influence to a measure that can be used to measure the influence of an internal neuron.

We first define a slice of a network. A particular layer in the network can be viewed as a slice. More generally, a slice is any partitioning of the network into two parts that exposes its internals. Formally, a slice $s$ of a network $f$ is a tuple of functions $\langle g, h \rangle$, such that $h : \mathcal{X} \to \mathcal{Z}$, and $g : \mathcal{Z} \to \mathbb{R}$, and $f = g \circ h$. The internal representation for an instance $\mathbf{x}$ is given by $\mathbf{z} = h(\mathbf{x})$. In our setting, elements of $\mathbf{z}$ can be viewed as the activations of neurons at a particular layer. The influence of an element $j$ in

the internal representation defined by $s = \langle g, h \rangle$ is given by

$$\chi_j^s(f, P) = \int_{\mathcal{X}} \frac{\partial g}{\partial z_j}\bigg|_{h(\mathbf{x})} P(\mathbf{x})d\mathbf{x} \tag{2}$$

We again take an axiomatic approach to justifying this measure, with two natural invariance properties on the structure of the network.

The first axiom states that the influence measure is agnostic to how a network is sliced, as long as the neuron with respect to which influence is measured is unchanged. Below, the notation $\mathbf{x}_{-i}$ referse to the vector $\mathbf{x}$ with element $i$ removed.

Two slices $s_1 = \langle g_1, h_1 \rangle$ and $s_2 = \langle g_2, h_2 \rangle$ are $j$-equivalent if for all $\mathbf{x} \in \mathcal{X}$, and $z_j \in \mathcal{Z}_j$, $h_1(\mathbf{x})_j = h_2(\mathbf{x})_j$, and $g_1(h_1(\mathbf{x})_{-j}z_j) = g_2(h_2(\mathbf{x})_{-j}z_j)$. Informally, two slices are $j$-equivalent as long as they have the same function for representing $z_j$, and the causal dependence of the outcome on $z$ is identical.

**Axiom 4** (Slice Invariance). *For all $j$-equivalent slices $s_1$ and $s_2$, $\chi_j^{s_1}(f, P) = \chi_j^{s_2}(f, P)$.*

The second axiom equates the input influence of an input with the internal influence of a perfect predictor of that input. Essentially, this encodes a consistency requirement between inputs and internal neurons that if an internal neuron has exactly the same behavior as an input, then the internal neuron should have the same influence as the input.

**Axiom 5** (Preprocessing). *Consider $h_i$ such that $P(X_i = h_i(X_{-i})) = 1$. Let $s = \langle f, h \rangle$, be such that $h(x_{-i}) = x_{-i}h_i(x_{-i})$, which is a slice of $f'(\mathbf{x}_{-i}) = f(\mathbf{x}_{-i}h_i(\mathbf{x}_{-i}))$, then $\chi_i(f, P) = \chi_i^s(f', P)$.*

We can now show that the only measure that satisfies these two properties is the one presented above (see Appendix C for the proof).

**Theorem 2.** *The only measure that satisfies slice invariance and preprocessing is Equation 2.*

## 3 EXPERIMENTS

In this section, we discuss some experiments that demonstrate the capabilities of our explanation framework. Our work generalizes other influence-based explanation frameworks, such as saliency maps (Simonyan et al., 2014) and integrated gradients (Sundararajan et al., 2017), meaning we have the ability to produce the same quality of explanations in the more-limited contexts in which these methods have been used. However, our explanation framework generalizes to axes left unexplored by these works, which can be used to provide richer explanations. In particular, we motivate the use of explanations for slices, various quantities of interest, and various distributions of interest.

We treat explanations as a means to answer specific queries about a model's behavior, and we show that the flexibility offered by our framework provides the tools necessary to confirm or refute specific hypotheses about a model's behavior.

### 3.1 EXPLANATIONS FROM SLICES

As discussed in Section 2.3, computing the influence on a slice of the network (Equation 2) lets us determine how relevant neurons in intermediate layers are to a particular network behavior. In particular, given an image and the network's prediction on that image, the influence measurements for a slice can reveal which features or concepts present in that image were relevant to the prediction.

Figure 1(a) shows the results of interpreting the influences taken on a slice of the VGG16 (Simonyan & Zisserman, 2014) corresponding to an intermediate convolutional layer (`conv4_1`). In this example we take the three most influential units for the quantity of interest corresponding to the correct classification of this image. More preciely, the quantity of interest used in this example corresponds to $f|_L$, i.e., the projection of the model's softmax output to the coordinate corresponding to the correct label $L$ of this instance. The visualization for each of these units was then obtained by measuring the influence of the input pixels on these units along each color channel, and scaling the pixels in the original image accordingly.

Because convolutional units have a limited receptive field, the resulting interpretation shows distinct regions in the original image, in this case corresponding to the left eye and mouth, that were most

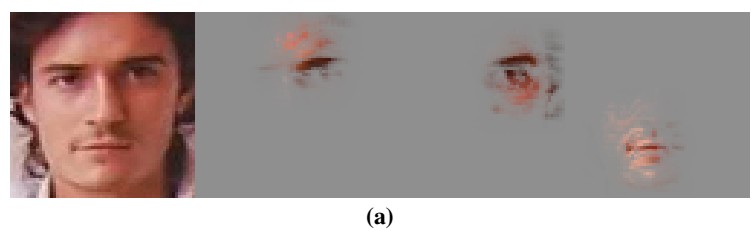 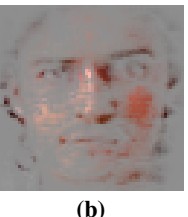

**(a)**                                       **(b)**

Figure 1: **(a)** Interpretation of the three most influential units from a slice corresponding to a convolutional layer (`conv4_1`), for the VGG16 (Simonyan & Zisserman, 2014) network. **(b)** Explanation based on integrated gradients (Sundararajan et al., 2017), taken on the same network and image. The interpretation in both cases was computed by scaling pixels in the original image using the results of either method.

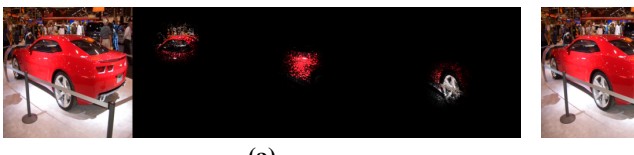 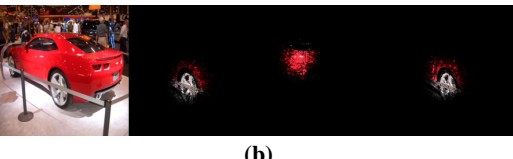

**(a)**                                       **(b)**

Figure 2: **(a)** Comparative explanation of classes "sports car" and "convertible" taken from the top-three most influential units at the `conv4_1` layer (VGG16 (Simonyan & Zisserman, 2014)). **(b)** Explanation computed using the quantity of interest corresponding to "sports car" on the same instance used in **(a)**.

relevant to the model's predicted classification. When compared to the explanation provided by integrated gradients (Sundararajan et al., 2017) shown in Figure 1(b), it is clear that the influence-directed explanation of the network's internal units is better at localizing the features used by the network in its prediction.

## 3.2 COMPARATIVE EXPLANATIONS

Influence-directed explanations are parameterized by a quantity of interest, corresponding to the function $f$ in Equation 1. Changing the quantity of interest affords flexibility in the characteristic explained by the influence measurements and interpretation. One class of quantities that is particularly useful in answering counterfactual questions such as, "Why was this instance classified as $L_1$ rather than $L_2$?", is given by the *comparative quantity of interest*.

More precisely, if $f$ is a softmax classification model that predicts classes $L_1, \ldots, L_n$, then let $f_{L_i}$ be the function $f_{L_i} = f|_i$ projected to the $i^{th}$ coordinate of its output. Then the comparative quantity of interest between classes $L_i$ and $L_j$ is $f_{L_i} - f_{L_j}$. When used in Equations 1 and 2, this quantity captures the tendency of the model to classify instances as $L_i$ over $L_j$.

Figure 2(a) shows an example of a comparitive explanation taken for a VGG16 (Simonyan & Zisserman, 2014) model trained on the ImageNet dataset (Russakovsky et al., 2015). The original instance shown on the left is labeled in the "sports car" leaf node of the ImageNet heirarchy. We measured influence using a comparative quantity against the leaf class "convertible", using a slice at the `conv4_1` convolutional layer. The interpretation was computed on the top-three most influential units at this layer in the same way as discussed in Section 3.1. The receptive field of the most influential unit corresponds to the region containing the hard top of the vehicle, which is understood to be its most distinctive feature from the convertible class. Figure 2(b) shows an interpretation for the same instance computed using influence measurements taken from the quantity of interest $f|_L$ (i.e., the same quantity used in Section 3.1 and implicitly used by integrated gradients (Sundararajan et al., 2017)). While both explanations capture features corresponding to common automobile features, only the comparative explanation isolates the portion that distinguishes "sports car" from "convertible".

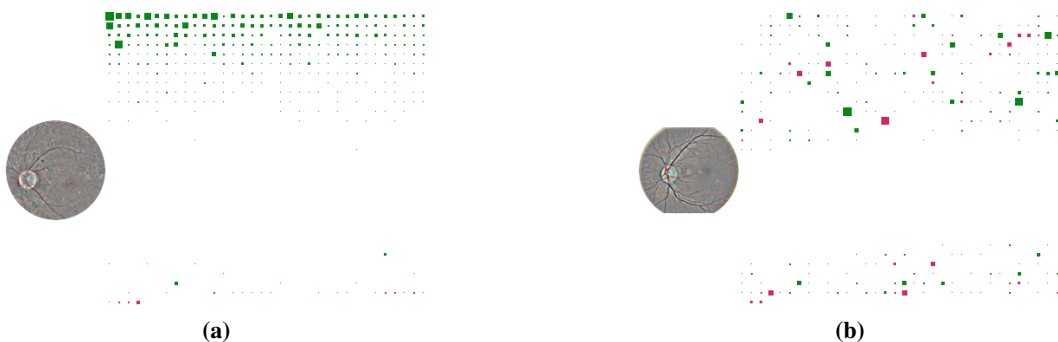

(a)                                                    (b)

Figure 3: Distributional influence measurements taken on DR model (Section 3.4) at bottom-most fully connected layer. To compute the grid, the distribution of influence was conditioned on class 5 (a) and class 1 (b). Figure (a) depicts an instance from class 5 that was correctly classified as such, and (b) an instance from class 5 that was incorrectly classified as class 1. In (a) the influences depicted in the grid align closely with the class-wide ordering of influences, whereas in (b) they are visibly more random. White space in the middle of the grid corresponds to units with no influence on the quantity.

## 3.3 DISTRIBUTIONS OF INTEREST

Another dimension along which influence-directed explanations can vary is the distribution $P$ used in Equations 1 and 2 to measure influence. The examples shown in Sections 3.1 and Sections 3.2 use a distribution of interest corresponding to a point mass defined over the single instance in question. The corresponding interpretations refer only to those instances, and so reflect specific features and concepts that may not generalize across a class. Defining the distribution of interest with support over a larger set of instances can yield explanations that capture the factors common to network behaviors across different populations of instances. These explanations capture the "essence" of what the network learned about a class, and may be useful in understanding problematic behaviors when debugging a network (see Section 3.4).

Figure 3 depicts an example of such an explanation. These visualizations were generated by measuring influence on a slice corresponding to the bottom-most fully-connected layer of the Inception Diabetic Retinopathy (DR) model (see Section 3.4 for details on this model). The quantity of interest $f|_i$ corresponds to a particular class outcome, and the distribution was conditioned on the corresponding class label. The units in that layer were then sorted according to their influence, with the top-left corner corresponding to the largest positive influence, and the bottom-right to the largest negative influence. For a specific instance (shown on the left of Figures 3(a) and (b)), influences at that layer were then measured, and the magnitude and sign of the corresponding unit in the class-wide ordering is depicted by the size and shape of the box at that position: large boxes denote larger magnitude, whereas green boxes denote positive sign and red negative.

Figure 3(a) depicts this for an instance of class 5 that was correctly classified, whereas 3(b) for an instance of the same class that was incorrectly classified. The difference is striking: for the correctly-classified instance, the influences align closely with the order determined by the distributional influence measurements, whereas they are noticeably more random in the incorrectly-classified case.

### 3.3.1 VALIDATING THE "ESSENCE" OF A CLASS

As is apparent in Figure 3, it is often the case that relatively few units are highly influential towards a particular class. In such cases, we refer to this as the "essence" of the class, as the network's behavior on these classes can be understood by focusing on these units. To validate this claim, we demonstrate that these units can be isolated from the rest of the model to extract a classifier that is more proficient at distinguishing class instances from the rest of the data distribution than the original model. To this end, we introduce a technique for compressing models using influence measurements to yield class-specific "expert" models that demonstrate the essence of that class learned by the model.

| Class | Recall (orig.) | Precision (orig.) | Recall (comp.) | Precision (comp.) |
|---|---|---|---|---|
| Chainsaw (491) | 14% | 100% | 71% | 100% |
| Bonnet (452) | 62% | 100% | 92% | 100% |
| Park Bench (703) | 52% | 100% | 71% | 100% |
| Sloth Bear (297) | 36% | 100% | 75% | 100% |
| Pelican (144) | 65% | 100% | 95% | 100% |

Figure 4: Model compression recall and accuracy for five randomly selected ImageNet classes.

Given a model $f$ with softmax output and slice $\langle g, h \rangle$ where $h : \mathbb{R}^d \to \mathcal{Y}$, let $M_h \in \mathbb{R}^d$ be a 0-1 vector. Intuitively, $M_h$ masks the set of units at layer $h$ that we wish to retain, and so is 1 at all locations corresponding to such units and 0 everywhere else. Then the *slice compression* $f_{M_h}(X) = g \circ h(X^T M_h)$ corresponds to the original model after discarding all units at $h$ not selected by $M_h$. Given a model $f$, we obtain a binary classifier $f^i$ for class $L_i$ (corresponding to softmax output $i$) by projecting the softmax output at $i$, in addition to the sum of all other outputs: $f^i = (f|_i, \sum_{j \neq i} f|_j)$.

**Class-specific experts.** For the sake of this discussion, we define a class-wise expert for $L_i$ to be a slice compression $f_{M_h}$ whose corresponding binary classifier $f^i_{M_h}$ achieves better recall on $L_i$ than the binary classifier $f^i$ obtained by $f$, while also achieving approximately the same recall. We demonstrate that the influence measurements taken at slice $\langle g, h \rangle$ over a distribution of interest $P_i$ conditioned on class $L_i$ yields an efficient heuristic for extracting experts from large networks.

In particular, we compute $M_h$ by measuring the slice influence (Equation 2) over $P_i$ using the quantity of interest $h|_i$. Given parameters $\alpha$ and $\beta$, we select $\alpha$ units at layer $h$ with the largest positive influence, and $\beta$ units with the greatest negative influence (i.e., greatest magnitude among those with negative influence). $M_h$ is then defined to be zero at all positions except those corresponding to these $\alpha + \beta$ units. In our experiments, we obtain concrete values for $\alpha$ and $\beta$ by a simple parameter sweep, ultimately selecting those values that yield the best experts by the criteria defined above. Figure 5 in the Appendix shows the precision and recall obtained for a range of $\alpha, \beta$, and notably that both measures plateau with relatively few units in the compressed model.

Figure 4 shows the precision and recall of experts found in this way for five randomly selected ImageNet classes, as well as the precision and recall of the original model on each class. We see that we find experts that have vastly better recall with no difference in precision. This suggests that the top and bottom influential neurons are truly sufficient to capture the essence of a class. The improvement we see in the compressed models seems to indicate that other neurons may capture spurious correlations in the training data that actually hurt the performance, while the essence of the class is a better indicator of class membership.

To further validate the hypothesis that influence measurements taken over a class-wide distribution of interest highlight specific components corresponding to the characteristics learned by the network, we repeated these experiments using activations rather than influences to compute $M_h$. On the same set of randomly-selected classes shown in Figure 4, the "experts" obtained in this way achieved negligible ($\approx 0$) recall, thus performing significantly worse than the original model. From this we see that considering activation levels alone does not provide the necessary information about the role of each unit in the network's behavior for these classes.

### 3.4 Case Study: Debugging a Diabetic Retinopathy Model

One of the primary ways that we envision influence-directed explanations being put to practical use is in debugging a model. In this section, we demonstrate a potential workflow for doing so, wherein a user poses queries about model behavior by crafting appropriate quantities and distributions of interest, and interpreting the resulting influence measurements to understand the reasons for incorrect model behavior. We show this in the context of Diabetic Retinopathy classification, which is a medical imaging task concerned with diagnosing the severity of a condition that affects the eye (Pratt et al., 2016; Gulshan et al., 2016). We replicate the results of prior work to build a model for this task, and construct a series of explanations that target questionable aspects of the model's behavior to discover the underlying cause of the problem.

**Background.**   We build on the prior work of Pratt et al. (Pratt et al., 2016) to build a convolutional network for predicting the severity of Diabetic Retinopathy (DR) in color retinal fundus images obtained from a Kaggle challenge dataset. DR is diagnosed on a scale from 1 to 5, with one corresponding to no symptoms of DR and 5 being the most severe presentation of symptoms. The dataset has a significant skew towards Class 1 with roughly 10 times as many images from Class 1 as compared to other classes. We were not able to come close to reproducing their results using the architecture presented in their report (Pratt et al., 2016). However, we found that using the Google Inception network (Szegedy et al., 2015) with the data preprocessing methods described by Pratt et al. (Pratt et al., 2016), the performance of our model matched the performance described by Pratt et al. with the overall accuracy within 1% of the results in (Pratt et al., 2016), and importantly, demonstrated the same pattern of misclassification as observed by Pratt et al. Namely, none of the images in the dataset are predicted as Severity-2 DR (mild symptoms), either correctly or incorrectly. The goal of our demonstration is to explain the cause of this behavior, which is not apparent at the outset.

**Explaining the bug.**   Given that no images are predicted as class 2, our initial exploration will attempt to determine whether the network did in fact learn concepts or features that distinguish class 2 from its neighbors. If the network did in fact learn concepts for class 2, it is possible that the bias in the distribution led the training process to prefer class 1 and 3 predictions over class 2, despite the presence of concepts associated with the latter in an instance. If the model were structured in a way that whenever units influential towards class 2 are activated, units associated with other classes, with more total influence on the network's output, are simultaneously activated, then we would expect to see the pattern of misclassifications present in this model. We call this the "drowning experts" hypothesis, and it would be characterized by the presence of experts (Section 3.3.1) for class 2 that are outweighed by experts for other classes. Under the assumption that the model contains an expert for class 2, we should be able to extract it using the techniques described in Section 3.3.1. Because we are particularly interested in positively identifying instances of class 2, we relax our definition of an expert to allow for those which increase recall for class 2, but possibly sacrifice some precision. Using the methods described in Section 3.3.1, we found such experts for every class *except* class 2, which suggests that our original suspicion that the model did not learn class 2 may in fact be correct.

Taking a look at the characteristics of the data, we observe that the features identifying classes 2-5 are lesions and other defects of increasing size, intensity, and frequency for higher-numbered classes. In practical terms, class 2 is distinguished from class 1 by very small features in the image space. However, as part of the preprocessing we applied to reproduce (Pratt et al., 2016), we apply a small Gaussian blur. This leads us to consider the possibility that the distinctive features of class 2 were in most cases erased from the corresponding instances, leaving the model with data labeled class 2 being indistinguishable from class 1. We confirm this hypothesis by removing the blur step from our preprocessing, and re-training the model. On evaluating the retrained model, the characteristic behavior on class 2 is not present, which is consistent with our hypothesis.

We conclude this section by noting that using a series of influence-directed explanations, we were able to diagnose and repair a curious problem in a complex model. By identifying portions of the model associated with certain predictive behaviors, we noted the absence of units responsible for predicting a particular class, which pointed us to a step in the data preprocessing phase of our training regime. Without the flexibility to define various quantities and distributions of interest, and to peer inside the model to understand the intermediate steps in the network's decisions, it is not clear that we would have been able to diagnose and repair this issue. Indeed, Pratt et al. (2016) hypothesized that the cause of the behavior was the limited depth of their network architecture; we observed the same phenomenon on a significantly larger network.

## 4   FUTURE WORK

We expect the distributional influence measure introduced in this paper to be applicable to a broad set of deep neural networks. One direction of future work is to couple this measure with appropriate interpretation methods to produce influence-directed explanations for other types of deep networks, such as recursive networks for text processing tasks. Another direction is to develop principled debugging support for models using influence-directed explanations as a building block.

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

## A    FEATURE COMPARISON TO PRIOR EXPLANATION FRAMEWORKS

Table 1 presents a comparison of the influence-directed explanation framework presented here to related prior work on explaining CNN decisions. The leftmost three columns describe framework properties for which some frameworks allow flexibility. For example, influence-directed and decomposition-based explanations can be computed for internal neurons in higher layers, as well as for neurons in the bottom-most input layer. In contrast, the frameworks proposed for Integrated Gradients (Sundararajan et al., 2017), Sensitivity Analysis (Simonyan et al., 2014), and the Simple Taylor Decomposition (Bach et al., 2015) are based on attributing relevance solely to the input features. Cells marked $\checkmark^*$ in these columns denote that the corresponding framework supports a limited amount of flexibility along the specified property.

| | Explanation framework properties | | | Influence measure properties | | |
| --- | --- | --- | --- | --- | --- | --- |
| | **Quantity** | **Distribution** | **Internal** | **Faithfulness** | **Sensitivity** | **Completeness** |
| influence-directed | $\checkmark$ | $\checkmark$ | $\checkmark$ | $\checkmark^*$ | $\checkmark$ | $\checkmark^*$ |
| integrated gradients | | $\checkmark^*$ | | $\checkmark^*$ | $\checkmark$ | $\checkmark$ |
| sensitivity analysis | | | | $\checkmark$ | | |
| deconvolution | | | $\checkmark$ | $\checkmark$ | | |
| guided backpropagation | | | $\checkmark$ | $\checkmark$ | | |
| relevance propogation | | | $\checkmark$ | $\checkmark$ | $\checkmark^*$ | $\checkmark$ |
| simple Taylor | | $\checkmark^*$ | | $\checkmark^*$ | $\checkmark$ | $\checkmark$ |

Table 1: Comparison of the influence-directed explanations proposed here to prior related work including Integrated Gradients (Sundararajan et al., 2017), Sensitivity Analysis (Simonyan et al., 2014), Deconvolution (Zeiler & Fergus, 2014), Layer-wise Relevance Propagation Bach et al. (2015), and Simple Taylor Decomposition (Bach et al., 2015). The first three columns refer to capabilities of the corresponding explanation framework: **Quantity** refers to flexibility in the choice of quantity to be explained; **Distribution** refers to flexibility in the distribution of instances to be explained; **Internal** refers to the ability to produce explanations that characterize the role of internal neurons. The latter three columns describe properties of the underlying influence measure used to build explanations: **Faithfulness** refers to distributional faithfulness; **Sensitivity** requires that if two instances differ in one feature and yield different predictions, then that feature is assigned non-zero influence; **Completeness** requires that the aggregate difference between influence on two instances sums to the difference of their outputs. See Sundararajan et al. (2017) for a more detailed discussion of Sensitivity and Completeness. Cells marked $\checkmark$ denote that the cited explanation framework has the corresponding feature, whereas those marked $\checkmark^*$ denote that the framework may have the feature under certain parameterizations.

The rightmost three columns in Table 1 refer to properties of the underlying influence measure used to construct explanations. Cells in these columns are marked $\checkmark^*$ if the framework satisfies the corresponding property under some parameterizations, but not necessarily in all of them. Notably, several of the frameworks make use of influence measures that do not satisfy sensitivity under any circumstances; this matter is described in further detail in (Sundararajan et al., 2017). Measures that do not satisfy sensitivity can be problematic in practice because they may fail to identify features or components that are causally-relevant to the quantity of interest, thus leading to "blind spots" or an inappropriate focus on irrelevant features.

Below we provide a detailed explanation for entries in Table 1 marked $\checkmark^*$.

**Influence-Directed.**    The influence-directed explanation framework presented in this paper supports all of the features in Table 1, but distributional faithfulness and completeness are contingent on the selection of appropriate parameters. Because our framework supports arbitrary distributions of

interest, there is no guarantee that the resulting explanation will reflect behaviors that are faithful to the training distribution. We view this as a feature, as it allows for explanations that may be relevant to counterfactual scenarios, and plan to explore the use of this feature in subsequent work. However, we note that in our framework ensuring distributional faithfulness is not difficult, and is achieved by using the source distribution, or a conditional or marginal variant of it. For example, the experiments in this paper use distributions corresponding to a single point from the training data, or the marginal distribution of instances in a given class.

As for completeness, many of the parameterizations of our framework have have this property. When the distribution of interest is chosen to coincide with that of integrated gradients, for example, then completeness follows immediately. We chose not to insist on completeness in all cases, as doing so would preclude the derivation of several useful classes of explanation. For example, explanations that exclude parts of the network to localize attribution may not satisfy completeness, but are useful in filtering out irrelevant information in explanations.

To make this concrete, observe that the explanations shown in Figures 1(a) and 2 (Section 3) were effectively computed by first identifying the most influential filters in an intermediate layer on a relevant quantity of interest, and then highlighting the input features that are influential on those intermediate features. The final attributions on the input features most likely do not sum to the quantity of interest because parts of the image excluded from the explanation are also relevant to the classification. Nonetheless, in both cases we obtain a useful explanation by ignoring certain attributions to identify localized features.

**Integrated Gradients.** Integrated gradients supports a limited form of flexibility on the distribution over which it computes attributions. Namely, by selecting a different baseline than the zero vector, the gradients will be aggregated over a different distribution of instances that may yield a qualitatively different explanation. However, integrated gradients can only include instances that are a linear combination of the baseline and input instances. The attributions given by integrated gradients remain plausibly faithful to the training distribution when the baseline is chosen appropriately, but establishing confidence in the faithfulness of a linear subspace defined by a particular baseline may not be an easy task. In the case of image models, when the black image is used as the baseline then attributions will be computed over variants of the image in question that differ in brightness. For models that operate over other types of data, the matter of distributional faithfulness needs to be considered on a case-by-case basis.

**Simple Taylor Decomposition.** The simple Taylor decomposition can be understood as a special case of integrated gradients only aggregates attributions for the baseline and input instance. Thus, like integrated gradients, it supports flexibility in the distribution of interest through calibration of the baseline. Distributional faithfulness also falls under the same conditions as for integrated gradients, and depends on the appropriate selection of a baseline instance.

**Relevance Propagation.** Layer-wise Relevance Propagation generalizes the propagation rule used in back-propagation methods. In particular, the generalization allows for any complete propagation rule to be used, thereby ensuring completeness. However, complete propagation rules do not satisfy sensitivity in all cases. For example, the quantity could be propagated through a single path of arbitrary neurons and still satisfy completeness. It is not clear that propagation rules not satisfying sensitivity can be considered causal. Thus, in order to capture causality, the propagation rule must be selected appropriately.

## B  Unique Measure

**Theorem 1.** *The only measure that satisfies linear agreement, distributional marginality and distribution linearity is given by Equation 1.*

*Proof.* Choose any function $f$ and $P_{\mathbf{a}}(\mathbf{x}) = \delta(\mathbf{x} - \mathbf{a})$, where $\delta$ is the Dirac delta function on $\mathcal{X}$. Now, choose $f'(\mathbf{x}) = \frac{\partial f}{\partial \mathbf{x}_i}|_{\mathbf{a}} x_i$. By linearity agreement, it must be the case that, $\chi(f', P_{\mathbf{a}}(\mathbf{x})) = \frac{\partial f}{\partial x_i}|_{\mathbf{a}}$. By distributional marginality, we therefore have that $\chi_i(f, P_{\mathbf{a}}) = \chi_i(f', P_{\mathbf{a}}) = \frac{\partial f}{\partial x_i}|_a$. Any distribution

$P$ can be written as $P(\mathbf{x}) = \int_{\mathcal{X}} P(\mathbf{a})P_{\mathbf{a}}(\mathbf{x})d\mathbf{a}$. Therefore, by the distribution linearity axiom, we have that $\chi(f, P) = \int_X P(\mathbf{a})\chi(f, P_a)da = \int_{\mathcal{X}} P(\mathbf{a})\frac{\partial f}{\partial x_i}|_{\mathbf{a}}d\mathbf{a}$. $\qquad \square$

## C  SLICE INVARIANCE

Two slices $s_1 = \langle g_1, h_1 \rangle$ and $s_2 = \langle g_2, h_2 \rangle$ are $j$-equivalent if for all $\mathbf{x} \in \mathcal{X}$, and $z_j \in \mathcal{Z}_j$, $h_1(\mathbf{x})_j = h_2(\mathbf{x})_j$, and $g_1(h_1(\mathbf{x})_{-j}z_j) = g_2(h_2(\mathbf{x})_{-j}z_j)$.

**Axiom 6** (Slice Invariance). *For all $j$-equivalent slices $s_1$ and $s_2$, $\chi_j^{s_1}(f, P) = \chi_j^{s_2}(f, P)$.*

**Axiom 7** (Preprocessing). *Consider $h_i$ such that $P(X_i = h_i(X_{-i})) = 1$. Let $s = \langle f, h \rangle$, be such that $h(x_{-i}) = x_{-i}h_i(x_{-i})$, which is a slice of $f'(\mathbf{x}_{-i}) = f(\mathbf{x}_{-i}h_i(\mathbf{x}_{-i}))$, then $\chi_i(f, P) = \chi_i^s(f', P)$.*

**Theorem 2.** *The only measure that satisfies slice invariance and preprocessing is*

$$\chi_j^s(f, P) = \int_{\mathcal{X}} \left.\frac{\partial g}{\partial z_j}\right|_{h(\mathbf{x})d\mathbf{x}}$$

*Proof.* Assume that two slices $s_1 = \langle g_1, h_1 \rangle$ and $s_2 = \langle g_2, h_2 \rangle$ are $j$-equivalent. Therefore, $g_1(h_1(\mathbf{x})_{-j}z_j) = g_2(h_2(\mathbf{x})_{-j}z_j)$. Taking partial derivatives with respect to $z_j$, we have that:

$$\left.\frac{\partial g_1}{\partial z_j}\right|_{h_1(\mathbf{x})_{-j}z_j} = \left.\frac{\partial g_2}{\partial z_j}\right|_{h_2(\mathbf{x})_{-j}z_j}$$

Now, since $h_1(\mathbf{x})_j = h_2(\mathbf{x})_j$, we have that

$$\left.\frac{\partial g_1}{\partial z_j}\right|_{h_1(\mathbf{x})} = \left.\frac{\partial g_2}{\partial z_j}\right|_{h_2(\mathbf{x})}$$

Plugging the derivatives into 6, we get that $\chi_j^{s_1}(f, P) = \chi_j^{s_2}(f, P)$. $\qquad \square$

where, for $P_{\mathcal{Z}}(\mathbf{z})$, we use the probability distribution induced by applying $h$ on $\mathbf{x}$, given by:

$$P_{\mathcal{Z}}(\mathbf{z}) = \int_{\mathcal{X}} P_{\mathcal{X}}(\mathbf{x})\delta(h(\mathbf{x}) - \mathbf{z})d\mathbf{x}.$$

Plugging this distribution into the influence measure above, we get:

$$\chi_j^s(g, P_{\mathcal{X}}) = \int_{\mathcal{Z}} \left.\frac{\partial g}{\partial z_j}\right|_{\mathbf{z}} P_{\mathcal{Z}}(\mathbf{z})d\mathbf{z} \tag{3}$$

$$= \int_{\mathcal{Z}} \left.\frac{\partial g}{\partial z_j}\right|_{\mathbf{z}} \int_{\mathcal{X}} P_{\mathcal{X}}(\mathbf{x})\delta(h(\mathbf{x}) - \mathbf{z})d\mathbf{x}d\mathbf{z} \tag{4}$$

$$= \int_{\mathcal{X}} P_{\mathcal{X}}(\mathbf{x}) \int_{\mathcal{Z}} \left.\frac{\partial g}{\partial z_j}\right|_{\mathbf{z}} \delta(h(\mathbf{x}) - \mathbf{z})d\mathbf{z}d\mathbf{x} \tag{5}$$

$$= \int_{\mathcal{X}} \left.\frac{\partial g}{\partial z_j}\right|_{h(\mathbf{x})} P_{\mathcal{X}}(\mathbf{x})d\mathbf{x}. \tag{6}$$

$$\tag{7}$$

Essentially, this shows that we can aggregate the partial derivates of the internal neurons on the distribution of interest over the input space itself. This measure also has an important slice invariance property that the influence on a neuron $z_i$ only depends on the functional relationship between $x$, $z_i$, and the outcomes. In other words, slices that differ only on other neurons will have the same influence for $z_i$. We formalize and prove this property in Appendix C.

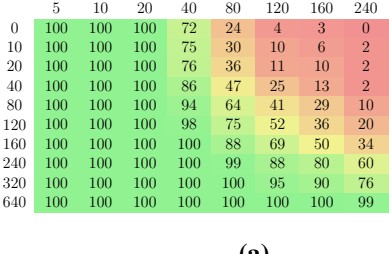 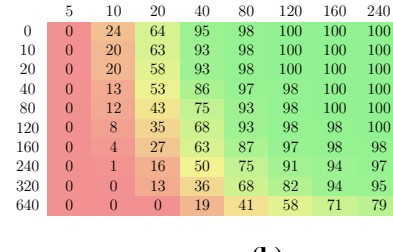

**(a)** **(b)**

Figure 5: (a) Precision and (b) recall for model compression results on a randomly-selected class from ImageNet using the VGG16 network. The rows correspond to $\alpha$, and columns to $\beta$. The results indicate that strong results can be obtained by selecting relatively few units for the compression mask $M_h$. While selecting larger sets of units does lead to increased performance, returns diminish rapidly.

## D    DISCUSSION ON INFLUENCE MEASURES

Influence measures are widely studied in cooperative game theory as solutions to the problem of attribution to of outcomes to participants and has applications to a wide range of settings including revenue division and voting. In this section, we highlight ideas drawn from this body of work and differences in terms of two key properties of influence measures: the *marginality principle*, and *efficiency*.

The *marginality principle* (Young, 1988) states that an agent's attribution only depends on its own contribution to the output. Formally, this is stated as: if the partial derivatives with respect to an agent of two functions are identical throughout, then they have identical attributions for agent $i$. Our axiom of distributional marginality (DM) is a weaker form of this requirement that only requires equality of attribution when partial derivatives are same in the distribution.

A second property, called *efficiency*, which is especially important for revenue division, is that attributions add up to the total value generated. This ensures that no value is left unattributed. The marginality principle, along with efficiency uniquely define the *Aumann-Shapley Value*(Aumann & Shapley, 1974). In Sundararajan et al. (2017), the Aumann-Shapley Value is used for attributions with efficiency as a justification. While it is unclear that efficiency is an essential requirement in our setting, the Aumann-Shapley value can be recovered in our framework by choosing the distribution of interest as the uniform distribution on the line segment joining an instance **x** and a baseline image **b**. Certain choices of baselines can be problematic from the point of view of distributional faithfulness, since the line segment of linear combinations between them might lie significantly out of distribution. The particular baseline chosen in Sundararajan et al. (2017) is the zero vector, where the line segment represents scaled images, and could be reasonably called within distribution.

