# OpenReview forum: "Influence-Directed Explanations for Deep Convolutional Networks"
_ICLR.cc/2018/Conference — Reject_

### Official Review · AnonReviewer1 · 2017-11-25
**Review for "Influence-Directed Explanations for Deep Convolutional Networks"**

**Rating:** 5
**Confidence:** 3

**Review:**

Notions of "influence" have become popular recently, and these notions try to understand how the output of a classifier or a learning algorithm is influenced by its training set. In this paper, the authors propose a way to measure influence that satisfies certain axioms. This notion of influence may be used to identify what part of the input is most influential for the output of a particular neuron in a deep neural network. Using a number of examples, the authors show that this notion of influence seems useful and may yield non-trivial results.

My main criticism of this paper is the definition of influence. It is easy to see that sometimes, influence of $x_i$ in a function $f(x_1, \dots, x_n)$ will turn out to be 0, simply because the integral in equation (1) is 0. However, this does not mean that $x_i$ is irrelevant to the the output f. This is not a desirable property for any notion of influence. A better definition would have been taking the absolute value of the partial derivative of f wrt x_i, or square of the same. This will ensure that equation (1) will always lead to a positive number as the influence, and 0 influence will indeed imply x_i is completely irrelevant to the output of f. These alternate notions do not satisfy Axiom 1, and possibly Axiom 5. But it is likely that tweaking the axioms will fix the issue. The authors should have at least mentioned why they preferred to use df/dx instead of |df/dx| or (df/dx)^2, since the latter clearly make more intuitive sense.

The examples in section 3 are quite thorough, but I feel the basic idea of measuring influence by equation (1) is not on solid footing.

---

### Official Review · AnonReviewer2 · 2017-11-27
**Comparison with other explanation methods missing**

**Rating:** 4
**Confidence:** 5

**Review:**

SUMMARY
========
This paper proposes to measure the "influence" of single neurons w.r.t. to a quantity of interest represented by another neuron, typically w.r.t. to an output neuron for a class of interest, by simply taking the gradient of the corresponding output neuron w.r.t to the considered neuron. This gradient is used as is, given a single input instance, or else, gradients are averaged over several input instances.
In the latter case the averaging is described by an ad-hoc distribution of interest P which is introduced in the definition of the influence measure, however in the present work only two types of averages are practically used: either the average is performed over all instances belonging to one class, or over all input instances.

In other words, standard gradient backpropagation values (or average of them) are used as a proxy to quantify the importance of neurons (these neurons being within hidden layers or at the input layer), and are intended to better explain the classification, or sometimes even misclassification, performed by the network.

The proposed importance measure is theoretically justified by stating a few properties (called axioms) an importance measure should generally verify, and then showing the proposed measure fullfills these requirements.

Empirically the proposed measure is used to inspect the classification of a few input instances, to extract "class-expert" neurons, and to find a preprocessing bug in one model. The only comparison to a related work method is done qualitatively on one image visualization, where the proposed method is compared to Integrated Gradients [Sundararajan et al. 2017].

WEAKNESSES
==========
The similarity and differences between the proposed method and related work is not made clear. For example, in the case of a single input instance, and when the quantity of interest is one output neuron corresponding to one class, the proposed measure is identical to the image-specific class saliency of [Simonyan et al. 2014].
The difference to Integrated Gradients [Sundararajan et al. 2017] at the end of Section 1.1 is also not clearly formulated: why is the constraint on distribution marginality weaker here ?
An important class of explanation methods, namely decomposition-based methods (e.g. LRP, Excitation Backprop, Deep Taylor Decomposition), are not mentioned. Recent work (Montavon et al., Digital Signal Processing, 2017), discusses the advantages of decomposition-based methods over gradient-based approaches. Thus, the authors should clearly state the advantages/disadvantes of the proposed gradient-based method over decomposition-based techniques.

Concerning the theoretical justification:
It is not clear how Axiom 2 ensures that the proposed measure only depends on points within the input data manifold. This is indeed an important issue, since otherwise the gradients in equation (1) might be averaged completely outside the data manifold and thus the influence measure be unrelated to the data and problem the neural network was trained on. Also the notation used in Axiom 5 is very confusing. Moreover it seems this axiom is even not used in the proof of Theorem 2.

Concerning the experiments:
The experimental setup, especially in Section 3.3.1, is not well defined: on which layer of the network is the mask applied? What is the "quantity of interest": shouldn't it be an output neuron value rather than h|i (as stated at the begin of the fourth paragraph of Section 3.3.1)?
The proposed method should to be quantitatively compared with other explanation techniques (e.g. by iteratively perturbing most relevant pixels and tracking the performance drop, see Samek et al., IEEE TNNLS, 2017).
The last example of explaining the bug is not very convincing, since the observation that class 2 distinctive features are very small in the image space, and thus might have been erased through gaussian blur, is not directly related to the influence measure and could have been made aso independently from it.

CONCLUSION
==========
Overall this work does not introduce any new importance measure for neurons, it merely formalizes the use of standard backpropagation gradients as influence measure.
Using gradients as importance measure was already done in previous work (e.g. [Simonyan et al. 2014]). Though taking the average of gradients over several input instances is new, this information might not be of great help for practical applications.
Recent work also showed that raw gradients are less informative than decomposition-based quantities to explain the classification decisions made by a neural network.

---

### Official Review · AnonReviewer3 · 2017-11-29
**An obvious extension of "use the gradient" for influence visualization**

**Rating:** 4
**Confidence:** 3

**Review:**

The authors extend traditional approach of examining the gradient in order to understand which features/units are the most relevant to given class.

Their extension proposes to measure the influence over a set of images by adding up influences over individual images. They also propose measuring influence for the classification decision restricted to two classes, by taking the difference of two class activations as the objective.

They provide an axiomatic treatment which shows that this gradient-based approach has desirable qualities.

Overall it's not clear what this paper adds to existing body of work:
1. axiomatic treatment takes a bulk of the paper, but does not motivate any significantly new method
2. from experimental evaluation it's not clear the results are better than existing work, ie Yosinsky http://yosinski.com/deepvis

---

### Decision · Program_Chairs · 2018-01-29
**ICLR 2018 Conference Acceptance Decision**

**Decision:**

Reject

**Comment:**

The paper defines a new measure of influence and uses it to highlight important features. The definition is novel however, the reviewers have concerns regarding its significance, novelty and a thorough empirical comparison to existing literature is missing.